

# Chemosensitivity and role of swimming legs of mud crab, *Scylla paramamosain,* in feeding activity as determined by electrocardiographic and behavioural observations

Gunzo Kawamura, Chi Keong Loke, Leong Seng Lim, Annita Seok Kian Yong and Saleem Mustafa

Borneo Marine Research Institute, Universiti Malaysia Sabah, Kota Kinabalu, Sabah, Malaysia

## ABSTRACT

Swimming crabs have a characteristic fifth pair of legs that are flattened into paddles for swimming purposes. The dactyl of these legs bears a thick seta along its edge. The chemoreceptive and feeding properties of the seta are supported with scientific evidence; however, there is no available data on the sensitivity of the setae in portunid crabs. The underlying mechanisms of the chemo- and mechano-sensitivity of appendages and their involvement in feeding activities of the mud crab (*Scylla paramamosain*) were investigated using electrocardiography and behavioural assay, which focused on the responses of the mud crab to chemical and touch stimulus. Electrocardiography revealed the sensory properties of the appendages. The dactyls of swimming legs and the antennules were chemosensitive, but not mechanosensitive and vice versa for the antennae. However, the mouthparts, claws, and walking legs were chemo- and mechanosensitive. Only the chemosensitive appendages, including the swimming legs, were directly involved in feeding. The flattened dactyls of the swimming legs were more efficient than the pointed dactyls of the walking legs in detecting the food organism crawling on the substrate. The structural features enhanced the capacity of the crab in coming into contact with scattered food items. This study revealed that the swimming legs are important appendages for feeding in the mud crab.

Corresponding author
Leong Seng Lim,
leongsen@ums.edu.my

## INTRODUCTION

Mud crabs, *Scylla* spp., are distributed in estuarine, sheltered coastal habitats, and mangroves in the Indo-Pacific region and South Africa (*Ikhwanuddin et al., 2011*; *Alberts-Hubatsch et al., 2016*). Four commercially important *Scylla* species (*S. serrata, S. olivacea, S. paramamosain,* and *S. tranquebarica*) live in the same geographical area in Malaysia (*Alberts-Hubatsch et al., 2016*) and are often captured in the same traps (*Kawamura et al., 2021*). In the wild population, inter-species mating has also been reported (*Fazhan et al., 2020*).

How to cite this article Kawamura G, Loke CK, Lim LS, Yong ASK, Mustafa S. 2021. Chemosensitivity and role of swimming legs of mud crab, *Scylla paramamosain,* in feeding activity as determined by electrocardiographic and behavioural observations. *PeerJ* **9**:e11248
http://doi.org/10.7717/peerj.11248

In crustaceans, all the appendages and several body parts are responsible for chemoreception (*Ache, 1982*). The most important chemosensory structures include the antennae, antennules (first antennae), walking legs, and mouthparts (*Schmidt & Mellon, 2011*). Several investigations have been conducted on lobsters (*Nishida et al., 1990; Gomez & Atema, 1996; Garm & Høeg, 2000; Garm, 2004a; Garm et al., 2005; Sahlmann, Chan & Chan, 2011*), crayfish (*Dunham & Oh, 1992; Garm, 2004b; Mellon Jr, 2012*), hermit crab (*Garm, 2004a; Kamio et al., 2005*), and blue crab *Callinectes sapidus* (*Gleeson, Adams & Smith, 1984; Keller, Powell & Weissburg, 2003; Aggio et al., 2012*). Nevertheless, *S. serrata* has been studied for the chemosensory function of its appendages amongst the *Scylla* spp. (*Wall, Paterson & Mohan, 2009*).

Decapod crustaceans possess setae on many parts of their appendages (*Ache, 1982*). To understand the behaviour of decapod crustaceans in the wild and under culture environments, it is important to study the sensory function of the appendages and their role in feeding and social communication. The setae could either be sensory or non-sensory. The former includes chemoreception or mechanoreception, which are determined using either one or a combination of behavioural, physiological, and morphological analysis. For studies based on behavioural assessment, an animal may have the capacity to perceive a stimulus, but it does not always lead to a change in response (*Nishi & Kawamura, 2005*). Physiological-based studies are limited as a result of non-detection of neural response, especially when the receptor's size is either small or labile (*Kawamura, 1981; Jordão, Cronin & Oliveira, 2007*), and when receptor potentials are unstable (*Kawamura et al., 2020*). The availability of different morphological types of setae among the decapod species makes the method uncertain while morphological analysis of setae provides a better understanding of the structures and functions (*Garm, 2004b*).

Electrocardiography is a reliable method of detecting the sensitivity of an animal to external stimuli (*Kawamura, Nakaizumi & Motohiro, 1992; Nishi, Kawamura & Matsumoto, 2004; Nishi & Kawamura, 2005*). In the tetrapods, the heart pacemaker is innervated by the excitatory sympathetic nervous system and the inhibitory parasympathetic nerve. However, in the heart of teleosts, the parasympathetic vagus nerve is the sole regulator of heart rate. Activation of the vagus nerve results in the release of acetylcholine–a neurotransmitter substance that acts via muscarinic receptors to influence the force and rate of contraction of the heart muscles. During resting or inactivity, inhibitory vagal stimulation on the heart is absent (*Taylor et al., 2007*). Fishes and crustaceans show a typical electrocardiogram response, slower heartbeat, or wider interbeat interval, in response to external stimuli. Electrocardiography has been applied to examine the response of crab species to temperature, pressure (*Mickel & Childress, 1982*), visual stimulus (*Grobel, 1990; Hermitte & Maldonado, 2006; Burnovicz, Oliva & Hermitte, 2009; Burnovicz & Hermitte, 2010*), and chemical stimulus (*Ketpadung & Tangkrockolan, 2006; Medesani et al., 2011*).

Mud crab (family Portunidae) is distinguished from other crabs by the structural adaptation of their fifth legs, which are transformed into flattened paddles and used for swimming purposes (*White & Sprito, 1973*). Hence, the structures are known as swimming legs. The swimming legs confer on the crabs a power to dart at high speed through water

bodies, while the dactyl bears thick setae along its edge. The chemoreceptive and feeding properties of the setae have been studied by different authors, however, there is no available data on the sensitivity of the setae in portunid crabs. This study investigated the chemo- and mechano-sensitivity of appendages and their involvement in feeding activities of the mud crab (*Scylla paramamosain*) using electrocardiography and behavioural assay. Both techniques were focused on the responses of the mud crab to chemical and touch stimulus.

## MATERIALS & METHODS

### Crabs and holding condition

A total of 20 female and male specimens of adult mud crab, *S. paramamosain,* (9–11 cm in carapace width, 80–100 g in body weight) caught from the wild population were obtained from a local market, Kota Kinabalu, Sabah. They were stocked in a plastic tank (122 cm ×148.5 cm, 55 cm high) filled with 55 L of seawater (30 cm deep) and placed in the roofed Shrimp Hatchery, Borneo Marine Research Institute, Universiti Malaysia Sabah. Half of the tank bottom was covered with 5 cm thick coral rubble to function as a biofilter substrate. Five pipes of polyvinyl chloride, 11 cm diameter and 20 cm long, were placed at the tank bottom as shelters for the crab. Tank water was aerated to ensure adequate oxygen level and the temperature, salinity, pH, and concentration of dissolved oxygen ranged from 25.8 to 28.9 °C, 19.47–20.05 ppt, 6.89–7.29, and 5.6–6.8 mg $L^{-1}$, respectively. These parameters were measured using a pH/ORD/EC/DO tester (Hanna Instruments, HI 9828, Washington, USA). The crabs were fed with fresh squid (*Loligo* sp.) in the morning and fish (*Decapterus* sp.) in the afternoon.

### Electrocardiography

Three crabs (one male and two females) were used for the electrocardiography experiment. The specimens were excitable in response to the movement of the experimenter, so their eyes were painted with white nail polish to temporarily block their vision. The crabs were cold-anesthetised followed by drilling a small hole of about 0.5 mm diameter at each cardiac lobe of the carapace to expose the pericardial membrane. Two active electrodes (enamel-coated copper wire, 0.05 mm diameter) were implanted through the holes and glued with cyanoacrylate. Crab specimens with implanted electrodes were left for a day to recover in separate tanks with well-aerated seawater at a temperature 28 °C, pH 8.1, and salinity 20 ppt.

To record the electrocardiogram (ECG), a test crab with two implanted electrodes was placed in a plastic test chamber (17 cm × 30 cm × 9.5 cm) filled with filtered seawater to 5 cm depth. The water level was enough to submerge the carapace. A third electrode was placed in the chamber as a ground reference. The recording wires were of sufficient length to allow the freedom of movement of the crab. However, the test specimens seldom moved during the ECG recordings. The test chamber water was kept at a temperature of 28 °C, pH of 8.1, and salinity of 20 ppt. The ECG was linked to an amplifier (MEG-1200, Nihon Kohden, Tokyo, Japan) aligned to a digital converter (FTT3P7PVB, ALS Controls, and Instruments, Nishinomiya, Japan), which converted analogue signals to digital signals. The signals were then processed with EasyLogger software (Easy SYNC Ltd., Glasgow, Scotland)
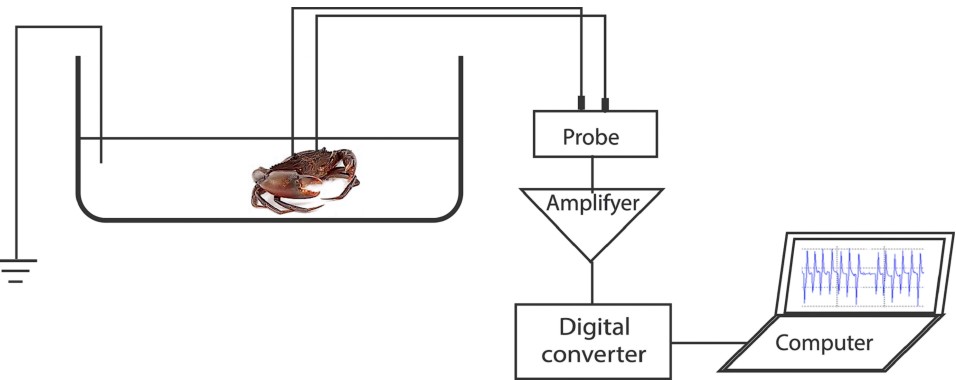

**Figure 1** Representation of the experimental setup used for recording the heart beat during the application of the different stimuli.

and displayed on a laptop monitor. Stable ECGs were obtained 15–30 min after the crab specimens were placed in the test chamber (Fig. 1). Subsequently, a stimulus (touch or sugarcane juice) was applied to the appendages. The ECGs were recorded during 25 s of stimulation.

## Stimuli

Each appendage of the crab was stimulated thrice in two ways (touch and chemical). The appendages were touched three times at intervals of 15–30 min with a hand-held micropipette (heat-pulled glass capillary with a 50 μm tip). For each crab, touch stimulation was conducted at the swimming leg dactyls, walking leg dactyls, claws, antennae (tip and middle part), antennules (lateral and medial flagella), and mouthpart (third maxilliped). Although the lateral and medial flagella are close to each other and frequently in motion, it was possible to touch them with the micropipette separately.

For the chemical stimulation, fresh sugarcane juice was selected because it contains 9.6–10.9% sucrose and 0.22–0.33% glucose and fructose, which are more effective feeding stimulants than amino acids for the fiddler crab *Uca pugilator* (*Robertson, Fudge & Vermeer, 1981*). The sugarcane juice measuring 0.1 mL (diluted to 50% or 10% with test chamber water, and filtered through paper Whatman No.1) was applied to the appendages by a hand-held Teflon pipette. Pipetting is one of the common methods to deliver a chemical solution to test crustaceans (*Cericola & Daniel, 2010*; *Aggio et al., 2012*). During the chemical stimulation, the threshold concentration of the sugarcane juice at the mouthparts and walking legs were maintained to comply with previous reports. According to *Liew et al. (2020)*, the threshold concentration of the sugarcane juice at the mouthparts and walking legs was between 1% and 10% in *S. tranquebarica*. The movement and dispersion of the sugarcane juice around the appendages were conspicuous due to its greenish colouration. Thereafter, the crab was removed and the test chamber was emptied. A control test was performed by replacing the sugarcane juice with the tank water (0.1 mL) before the application of the sugarcane juice, which was presented in triplicates.

To prevent the diffusion of the sugarcane juice from the antennule to the antennae, the antennules were cauterised using an electric cauterizer (Gemini BC00314, Cellpoin Scientific, Gaithersburg, Germany) after applying the sugarcane juice. Thereafter, the sugarcane juice was applied to the antennae. For the cauterization, the crab was taken out of the test chamber, cold-anesthetised, and gently held with a wet towel, and the tip of the cauterizer was touched on the distal segment bearing the outer and inner flagellum. After the cauterisation, the crab was placed back into the chamber and left for 20 to 30 min to recover to its stable and uniform ECG. The antennae were stimulated with 10% and 50% sugarcane juice. Each stimulation was delivered at an interval of 20–30 min to minimise residual responses to a previous stimulus and to avoid desensitization.

**Recording and measuring electrocardiograms**

Fifty-four stable ECGs were obtained from three test crabs: constant interbeat intervals (IBIs) of $0.5 - 0.8$ s and constant amplitude of $0.3 - 0.4$ mV. During the stimulus application, the ECGs were examined for the IBIs, measured to 0.1 s. The first post-stimulus IBI was considered as the relevant response. The IBI values were normalised by logarithmic transformation (*Nishi, Kawamura & Matsumoto, 2004*): normalised interval$= \log_{10}(T+1)$, where $T$ is the raw value of the IBI (s). Each test IBI was then compared with the mean pre-test IBI (the test-beat is out of the 95% confidence interval calculated using 10 pre-test IBIs).

**Behavioural bioassay**

During the stimulation at each appendage, the motion of other appendages was observed visually and combined with video recordings using CCD cameras with temporal resolutions of 0.017 s, corresponding to 60 frames per second (Olympus Tough TG-3, Olympus Corporation, Tokyo, Japan). A total of 63 video recordings were analyzed, which were played back to examine the feeding behaviour on a computer screen.

After the electrocardiography experiment, the feeding behaviour of each crab was observed in the test chamber. The antennule was stimulated with a tweezered small piece of salted fish flesh (*Decapterus* sp.) (about 10 mm ×5 mm, two mm thick) obtained from a local market. To stimulate the walking legs or swimming legs, a small piece of salted fish flesh was placed close to their dactyl. As described by *Liew et al. (2020)*, the feeding behaviour in response to the stimulation was evaluated based on three appendage movements: (i) increase in antennular flicking rate, (ii) remarkable movement of the third maxillipeds, and (iii) probing the test chamber bottom with the claws. These behaviours were visually observed, video-recorded, and a total of 30 video recordings were analyzed.

## RESULTS

### Electrocardiography

The electrocardiogram showed typical heartbeat response, slower heartbeat, or wider IBI of the crab, after the 10% or 50% sugarcane juice stimulation was presented to the swimming leg dactyl (Fig. 2A). However, no response was observed by mechanical touch stimulation (Fig. 2B). The crab with the antennules showed the heartbeat response, while those with

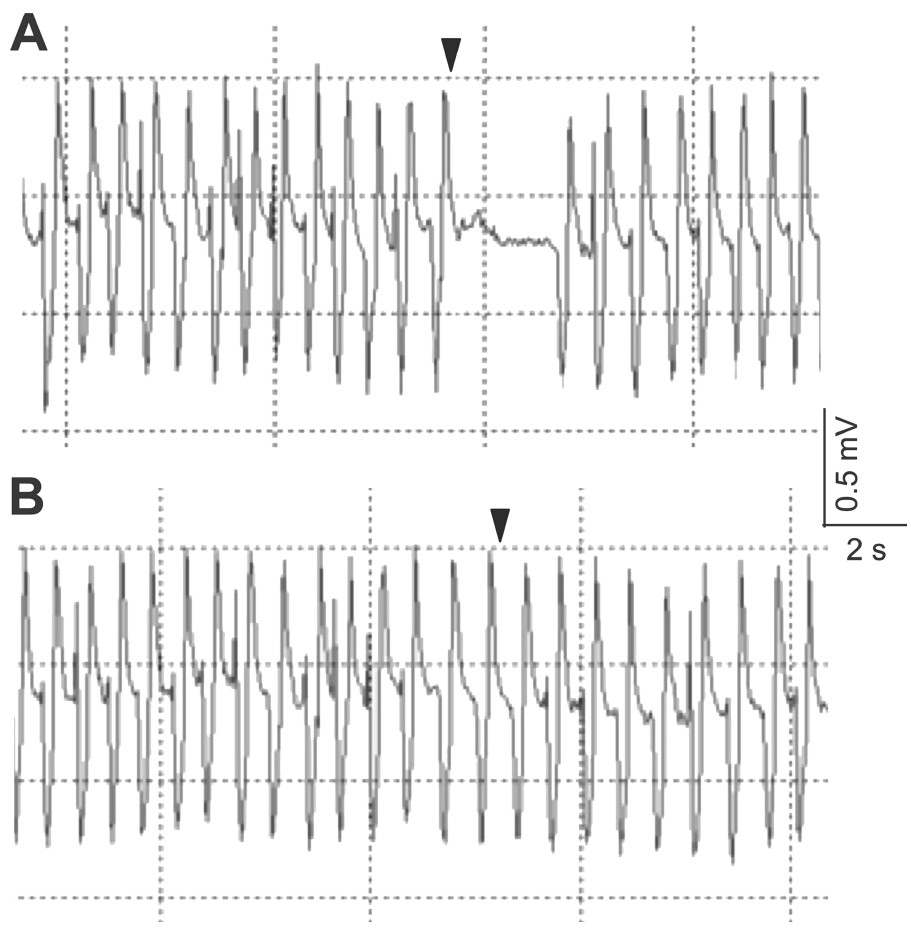

**Figure 2** **Typical electrocardiograms showing the heart beat response of *Scylla paramamosain* after the presentation of 50% sugarcane juice (A) and no change in heart beat interval in response to a single touch stimulus (B) delivered to the swimming leg dactyl.** Arrows represent the application of the stimulus.

the cauterised antennules showed no response to the stimulation by the sugarcane juice application given to the antennae. A statistically significant ($P < 0.05$) cardiac response to the sugarcane juice was recorded for the swimming legs, walking legs, claws, antennules (both the lateral and medial flagella), and mouthparts (Table 1). Regarding the touch test, statistically significant responses ($P < 0.05$) were observed for the walking legs, claws, antennae, and mouthparts of three specimens (Table 2). Both the lateral and medial flagella were not touch-sensitive. In the control test, the crab exhibited neither the cardiac response nor the feeding behaviour in response to tank water that was applied to all appendages.

## Behaviour change in response to stimuli

Stimulation with the sugarcane juice elicited both the cardiac and feeding response of the crab. During electrocardiography, the mud crab exhibited an increase in the antennule flicking rate, claw probing, and movement of the third maxillipeds. These responses occurred following the stimulation of the antennules and dactyls (both swimming and

**Table 1 Change in interbeat intervals (IBI) in *Scylla paramamosain* in response to sugarcane juice delivered to appendages.**

| Appendage | Crab | Sugarcane juice concentration (%) | Pre-test IBI: log10-transformed mean IBI (s) | Pre-test IBI: log10-transformed IBI 95% confidence interval (s) | Test-beat: log10-transformed IBI (s) |
|---|---|---|---|---|---|
| Swimming leg | A (male) | 10 | 0.248 | 0.171−0.325 | 0.646[*] |
| | | 50 | 0.237 | 0.169−0.305 | 0.898[*] |
| | B (female) | 10 | 0.290 | 0.249−0.331 | 0.643[*] |
| | | 50 | 0.266 | 0.233−0.299 | 0.981[*] |
| | C (female) | 10 | 0.237 | 0.207−0.267 | 0.620[*] |
| | | 50 | 0.247 | 0.202−0.292 | 0.929[*] |
| Walking leg | A | 10 | 0.404 | 0.350−0.458 | 0.737[*] |
| | | 50 | 0.261 | 0.206−0.316 | 0.729[*] |
| | B | 10 | 0.331 | 0.261−0.401 | 0.659[*] |
| | | 50 | 0.246 | 0.178−0.314 | 0.676[*] |
| | C | 10 | 0.400 | 0.354−0.446 | 0.898[*] |
| | | 50 | 0.203 | 0.168−0.238 | 0.484[*] |
| Claw | A | 10 | 0.273 | 0.223−0.313 | 0.979[*] |
| | | 50 | 0.243 | 0.196−0.290 | 0.587[*] |
| | B | 10 | 0.250 | 0.204−0.296 | 0.754[*] |
| | | 50 | 0.276 | 0.234−0.318 | 1.157[*] |
| | C | 10 | 0.261 | 0.194−0.328 | 0.618[*] |
| | | 50 | 0.285 | 0.109−0.461 | 0.771[*] |
| Antenna | A | 10 | 0.216 | 0.188−0.244 | 0.236 |
| | | 50 | 0.137 | 0.066−0.208 | 0.090 |
| | B | 10 | 0.180 | 0.150−0.210 | 0.207 |
| | | 50 | 0.089 | 0.020−0.158 | 0.117 |
| | C | 10 | 0.246 | 0.206−0.286 | 0.217 |
| | | 50 | 0.198 | 0.127−0.269 | 0.196 |
| Antennule | A | 10 | 0.223 | 0.169−0.277 | 0.633[*] |
| | | 50 | 0.284 | 0.241−0.327 | 0.844[*] |
| | B | 10 | 0.187 | 0.136−0.238 | 0.375[*] |
| | | 50 | 0.253 | 0.201−0.305 | 0.832[*] |
| | C | 10 | 0.255 | 0.173−0.337 | 0.862[*] |
| | | 50 | 0.180 | 0.144−0.216 | 1.097[*] |
| Mouthparts | A | 10 | 0.308 | 0.261−0.355 | 0.671[*] |
| | | 50 | 0.339 | 0.307−0.371 | 0.969[*] |
| | B | 10 | 0.301 | 0.265−0.337 | 0.746[*] |
| | | 50 | 0.306 | 0.285−0.327 | 0.859[*] |
| | C | 10 | 0.277 | 0.224−0.310 | 0.633[*] |
| | | 50 | 0.339 | 0.280−0.398 | 0.754[*] |

**Notes.**

The stimulation was triplicated but cardiac responses were shown only for the first single touch. Mean pre-test IBI was calculated for 10 pre-test IBI.

*denotes significantly larger test-beat interval than 95% confidence interval calculated using 10 pre-test IBIs.

**Table 2** Change in interbeat intervals (IBI) in *Scylla paramamosain* in response to touch stimulus delivered to appendages.

| Appendage | Crab | Pre-test IBI: log10-transformed mean IBI (s) | Pre-test beat: log10-transformed IBI 95% confidence interval (s) | Test-beat: log10-transformed IBI (s) |
|---|---|---|---|---|
| Swimming leg | A (male) | 0.139 | 0.063–215 | 0.097 |
| | B (female) | 0.170 | 0.125–215 | 0.146 |
| | C (female) | 0.210 | 0.181–239 | 0.193 |
| Walking leg | A | 0.337 | 0.287–387 | 0.782* |
| | B | 0.307 | 0.237–377 | 0.646* |
| | C | 0.329 | 0.287–371 | 0.868* |
| Claw | A | 0.323 | 0.264–382 | 0.627* |
| | B | 0.330 | 0.316–344 | 0.658* |
| | C | 0.359 | 0.326–392 | 0.718* |
| Antenna | A | 0.226 | 0.163–289 | 0.679* |
| | B | 0.244 | 0.219–269 | 0.668* |
| | C | 0.245 | 0.213–277 | 0.765* |
| Antennule | A | 0.139 | 0.063–215 | 0.124 |
| | B | 0.244 | 0.197–251 | 0.220 |
| | C | 0.184 | 0.115–253 | 0.225 |
| Mouthparts | A | 0.251 | 0.205–297 | 0.684* |
| | B | 0.319 | 0.289–349 | 0.647* |
| | C | 0.154 | 0.126–182 | 0.563* |

**Notes.**
The stimulation was triplicated but cardiac responses were shown only for the first single touch. Mean pre-test IBI was calculated for 10 pre-test IBI.
*Denotes significantly larger test-beat interval than 95% confidence interval calculated using 10 pre-test IBIs.

walking) with sugarcane juice. Such feeding behaviour was not observed since the antennae were touch-sensitive (Table 2), but not chemosensitive (Table 1) . The touch stimulus alone did not elicit the feeding behaviour at all.

In response to a small piece of salted fish flesh placed on the swimming leg dactyl, the mud crab immediately exhibited movement of mouthpart appendages and kicked the flesh to the claw under its body (Fig. 3). Other exhibited features include pushing the flesh to the walking leg dactyl and grabbing the piece with the claw to bring it to the mouth. There was a clear involvement of the swimming legs in feeding.

## DISCUSSION

The present study reinstated the importance of swimming legs of the mud crab for swimming and its vital role in food detection. The dactyl of swimming legs was found to be chemosensitive and when the dactyl touched a piece of food, the food was quickly kicked forward under the body by the motion of swimming legs, grabbed by the claw, and ingested. The kicking motion was done without touch sensitivity. Among the legs in the mud crab, the swimming legs are farthest from the mouth, nevertheless, the motion of the swimming leg secures food grabbing by the claw.

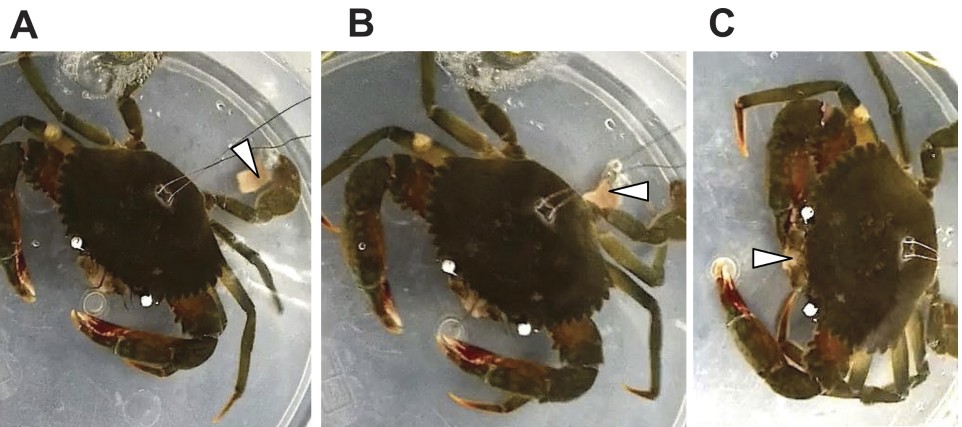

**Figure 3** **Sequential images of a food capture process with the swimming leg of *Scylla paramamosain*.**
(A) Touching the dactyl of the swimming legs by a piece of fish flesh (arrow head); (B) kicking the fish flesh under the body toward the mouth; (C) grabbing the fish flesh with a claw and conveying it to the mouth.

Mud crab is a benthic predator, an opportunistic scavenger, and feeds on sessile or slow-moving benthic macroinvertebrates, mainly gastropods, crustaceans, and molluscs (*Alberts-Hubatsch et al., 2016*). It also consumes fresh and decaying flesh of all kinds, hence, regarded as a detritivorous animal (*Mamun et al., 2008*; *Nesakumari & Thirunavukkarasu, 2014*; *Viswanathan & Raffi, 2015*; *Paul et al., 2018*). Previous studies have shown that *S. serrata* uses the dactyls of the walking legs by chemoreception to locate food (*Alberts-Hubatsch et al., 2016*). Besides, the flattened dactyls of the swimming legs might be more efficient than the pointed dactyls of the walking legs during food location. When crawling on the substrate, the dactyls of the swimming legs sweep the surface, which increases the likelihood of contacting scattered food items compared with the walking legs. Thus, the swimming legs of the mud crab are the most suited more appendage in locating food.

The swimming legs are also used in burying behaviour (*Parkes, Quinitio & Vay, 2011*). In aquaria, a pair of swimming legs are applied to flick sediment over the anterior of the carapace in *S. serrata* (*Parkes, Quinitio & Vay, 2011*). To achieve this burying motion, the mud crab requires information about the quality of bottom sediment. Since the dactyl of the swimming legs is not touch-sensitive, the cue of the sediment quality might be mediated by the other mechanosensitive appendages such as the walking legs. According to *Fedetov (2009)*, the hair receptors on chelipeds, antennas, and antennules in crayfish and other decapods are innervated by mechano- and chemoreceptor neurons and bimodal sensillae. In the present study, it was evident that the short antenna of the mud crab, unexpectedly, was touch-sensitive but not chemosensitive to food-related organic substances. In contrast, the antennae have been reported to be used mechanically in feed deposit and suspension in decapod crustaceans (*Boxshall & Jaume, 2013*). This event suggests that the antennae are not directly involved in food detection. However, the potential role of the antennae in detecting the source of food odour is not ruled out. *Weissburg (1994)* reported that the blue crab could not orient the source of food odour in still water, but proceeded directly

upstream toward the food odour source. Their findings indicated the importance of both rheotactic and chemical information for successful orientation.

Mechanoreceptors mediate remote perception of hydrodynamic signals such as disturbances in flow field caused by particles suspended in water columns or by the motions of live prey (*Boxshall & Jaume, 2013*). Assuming that the touch-sensitive antennae mediate hydrodynamic signals, the mud crab would be able to orient food odour source by olfaction aided by the perception of hydrodynamic signals by the antennae. The orientation may also be enhanced by the monomodal touch-sensitive sensillae reportedly present on the bristle patches on the carapace of *Scylla* spp. (*Kawamura et al., 2021*). The bristle patches are located behind the eyes in both males and females of four species of the mud crabs, and a touch stimulus given to the bristles is a mechanical signal for their courtship behaviour.

The antennule plays an important role in many behaviours of the mud crab, including locomotion, feeding, and mating (*Boxshall & Jaume, 2013*). The lateral flagellum of the antennule bears bimodal sensilla, which are innervated by mechanoreceptor and chemoreceptor neurons in decapod crustaceans (*Schmidt & Mellon, 2011*). The bimodal sensitivity of the antennule has been reported in several species of lobsters and shrimps. However, in the present study, the lateral flagellum of the antennule of the mud crab is a unimodal chemosensor, distinctly chemosensitive, but not touch-sensitive. *Van Weel & Christofferson (1966)* conducted an electrophysiological study of appendages of two portunid crabs: *Podpphthalmus vigil* and *Portunus sanguinolentus*. In their experiment, touch stimulation of walking legs, antennae, and mouthparts all caused the electroactivity in the antennule. This is possibly due to the uncommon arrangement of a reference electrode, which was inserted into the carapace and was not grounded. A reference electrode is typically connected to a grounded electrode, which is essential for most electrophysiological recordings. Therefore, the results of *Van Weel & Christofferson (1966)* are questionable so far as this aspect is concerned. The present study revealed the bimodal sensitivity of the mouthparts, claws, and walking legs of the mud crab, which seemed to be inconsistent with those of other decapod crustaceans based on available literature findings.

## CONCLUSIONS

Electrocardiography revealed that the thick setae along the edge of the swimming leg dactyl of the mud crab, *Scylla paramamosain,* were chemosensitive but not mechanosensitive, whereas the antennae were mechanosensitive but not chemosensitive. Behavioural observations showed the involvement of swimming legs in feeding. A combination of electrocardiographic and behavioural patterns of appendages indicated that the body parts directly involved in feeding include the mouthparts, claws, walking legs, swimming legs, and antennules.

### Funding

This work was supported by the research grant (No. SDK0028-2018) provided by the Universiti Malaysia Sabah. There was no additional external funding received for this study. The funders had no role in study design, data collection and analysis, decision to publish, or preparation of the manuscript.

### Grant Disclosures

The following grant information was disclosed by the authors:
Universiti Malaysia Sabah: SDK0028-2018.

### Competing Interests

The authors declare there are no competing interests.

### Author Contributions

- Gunzo Kawamura conceived and designed the experiments, performed the experiments, analyzed the data, prepared figures and/or tables, authored or reviewed drafts of the paper, and approved the final draft.
- Chi Keong Loke performed the experiments, prepared figures and/or tables, and approved the final draft.
- Leong Seng Lim conceived and designed the experiments, performed the experiments, analyzed the data, authored or reviewed drafts of the paper, and approved the final draft.
- Annita Seok Kian Yong conceived and designed the experiments, performed the experiments, analyzed the data, authored or reviewed drafts of the paper, and approved the final draft.
- Saleem Mustafa performed the experiments, analyzed the data, authored or reviewed drafts of the paper, and approved the final draft.

### Data Availability

The data is available at figshare: Lim, Leong Seng; Kawamura, Gunzo (2020): Swimming legs are chemosensitive and involved in feeding in the mud crab Scylla paramamosain as determined electrocardiography and behaviourally. figshare. Dataset. https://doi.org/10.6084/m9.figshare.13012799.v2.

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
