# Peer review of "Chemosensitivity and role of swimming legs of mud crab, Scylla paramamosain, in feeding activity as determined by electrocardiographic and behavioural observations"

_PeerJ, doi:10.7717/peerj.11248_

## Round 0.1 · original submission · Major Revisions

Your manuscript has to be deeply improved and modified. The real interest of the study should be included in the introduction in order to make the paper more attractive. Tables 1 and 2 lack details and it is mandatory to include in them the number of samples, mean of the value, and statistic results. More comments have been provided by the reviewers. I invite you to revise and resubmit your manuscript.

Reviewer 1 ·

Basic reporting

This manuscript is written in understandable English.

Sufficient back ground is provided.

Article structure and figures are fine.

Table 1 and 2 lack details, number of samples, mean of the value.

Average and standard error of pretest and test should be shown in the table or in a graph.

Behavioral result is qualitative but not quantitative. Number of trials and number of observations of behavior of interest should be provided.

I did not find raw data.

The manuscript is self-contained.

Experimental design

This is an original primary research within Aims and Scopes of PeerJ.

Main research question is simple, "Do dactyl of swimming leg have chemo sensory function in the green mud crab?" and well defined. Since the swimming leg is one of pleopods that had been reported as chemosensory organ and thus, the question is not so novel. However, testing swimming leg in this species is novel.

The experiments were performed rigorously and to a high technical standard. Ethical issue is fine.

Method description is fine.

Validity of the findings

Quantitative information of behavioral experiments (number of trial and observation and statistical test) are missing.

Main conclusion, "Dactyl of swimming leg has chemo sensory function in the green mud crab." is fine, however, the experiments in this research can not conclude that a part of bodies tested does not have mechano- or chemoreceptor. The part should be rewritten.

Additional comments

The manuscript entitled “Swimming legs are chemosensitive and involved in feeding in the mud crab Scylla paramamosain as determined electrocardiography and behaviourally (#51510)” reports that the dactyls of the swimming legs of the mud crab have chemical sense. This finding is new and supported by the data presented. Electrocardiography is a reliable method when it shows positive response but we can not conclude that “the dactyls of swimming legs and the antennules are not mechanosensitive; the antennae are not chemosensitive based on the result. It is known that the antennule has mechanoreceptors in other decapods (ex. Kamio and Derby 2017). This is partially because electrocardiography is not direct measurement of chemosensory input in chemosensory neurons. It is a recording of changes in heart beat that is caused by input from the central nerve system that integrates multi-sensory inputs. Thus, negative data in electrocardiography does not mean that the crab is not sensing the stimuli. There might be sensory inputs that does not make changes in electrocardiograph. Overall, this manuscript has novel finding but need to correct overstatements. I recommend authors to rationalize the method, electrocardiography, explaining not only its strength but also its weakness. Behavioral result racks quantitative data. Please show data as table or graph. Please show number of crabs showed and did not show the response. I think this manuscript is publishable after revision.

Kamio, M. and C. D. Derby (2017). "Finding food: how marine invertebrates use chemical cues to track and select food." Natural Product Reports 34(5): 514-528.

Annotated reviews are not available for download in order to protect the identity of reviewers who chose to remain anonymous.

Reviewer 2 ·

Basic reporting

The study characterises the sensory function of the appendages which is important and fundamental in understanding the feeding behaviour of mud crab. However, there is no clear problem or problem statement that emphasise on the needs or importance to perform this study, that is to determine the sensory function of appendages, hence making the study less interesting, and merely just characterising. It would be interesting to relate the problem of feeding of mud crab in nature or in aquaculture. For example slow eating behaviour of mud crab leads to nutrient leaching of feed that deteriorates water quality, hence reducing feed efficiency and increasing feed cost. By understanding the sensory function and feeding behaviour of mud crab in this study, feeds and feeding of mud crab can be optimized to cater its natural feeding physiology and behaviour. English is acceptable but there are some grammatical/typing errors that needs to be rechecked. Include more literature on the importance of the appendages and its sensitivity in feeding or daily activities. Also elaborate more on the fundamental concept of the methodology used especially on the electrocardiography and sensitivity so that readers have better understanding on the trend of the graph (electrocardiogram). The manuscript also lacks hypothesis in order to justify the results.

Experimental design

Experimental design should be explained in detail especially on the video recording of the feeding behaviour. What kind of feeding behaviour that was analysed? The chronology of procedure for stimulation of crab using touch and chemical is a little confusing. Was the cauterisation of antennules performed after touch stimuli and before chemical stimuli, OR even before the touch stimuli. Please be clear. Authors mentioned that all procedures were conducted according to the approved ethical standards, but was the cauterisation performed without any anaesthetic? Anatomy of mud crab showing the appendages will be helpful. At the current state of the manuscript, the only knowledge gap is that this study was conducted in some other crustacean species but has not been performed in S. paramamosain. As mentioned in the basic reporting, there is no clear problem or problem statement of the study, therefore the contribution towards filling the knowledge gap is not clear as well. In general, parameters studied in this work is insufficient as an original paper. Perhaps it is more suitable for short communication.

Validity of the findings

Authors claimed that the swimming legs were chemosensitive to sugarcane juice and salted fish, and the kicking motion was not triggered with a single touch stimuli. Just a thought, can it be that the single touch was just in a short moment (just a touch) and insufficient time to trigger a movement/reaction, while the sugarcane juice spreads throughout the water volume and stays for longer period of time, enough to trigger the chemoresceptor that is not from the swimming leg? Similar to the salted fish flesh in regards to the chemosense, with the addition to being placed at the swimming leg long enough to trigger a reaction compared to the single touch stimuli, hence may be related to the flaw of experimental design? Possibly, an object without smell should be placed similarly as the salted fish flesh and clear water should be delivered at the appendages using a pipette such as for the sugarcane juice in order to omit the possibility of rheotaxis effect. How confident are the authors to stand by the claims should be discussed in detail with supportive references since not much data were obtained from this study.

Additional comments

Most comment were covered by the 3 areas above.

Annotated reviews are not available for download in order to protect the identity of reviewers who chose to remain anonymous.

---

## Round 0.2 · Minor Revisions

It is necessary and important to make minor changes in the manuscript. The results should be rewritten (see comments of reviewer one for more information).

Reviewer 1 ·

Basic reporting

Description of results need to be modified.

(1) Result of behavioral experiment is only qualitative but not quantitative. Number of trials and number of occurrence of feeding behavior should be provided in the manuscript.


(2) Line 178 needs to be rewrite because T-test is not applicable on this data. This case compares 10 pre-test IBIs and a Test-beat (see table 1&2) from an animal. Thus, in table 1 and 2, * denotes that the Test-beat is out of 95% confidence interval calculated using 10 pre-test IBIs.

Experimental design

No comment

Validity of the findings

Description of results need to be modified.

(1) Result of behavioral experiment is only qualitative but not quantitative. Number of trials and number of occurrence of feeding behavior should be provided in the manuscript.


(2) Line 178 needs to be rewrite because T-test is not applicable on this data. This case compares 10 pre-test IBIs and a Test-beat (see table 1&2) from an animal. Thus, in table 1 and 2, * denotes that the Test-beat is out of 95% confidence interval calculated using 10 pre-test IBIs.

Reviewer 2 ·

Basic reporting

Authors have made the necessary changes on the manuscript.

Experimental design

The experimental design was clearly explained and aligned with the objective of the study.

Validity of the findings

Findings are well discussed and concluded.

Additional comments

No comment.

---

## Round 0.3 · Minor Revisions

Thank you very much for improving your manuscript. Although the scientific content of the submission is ready, the paper needs to be edited by a fluent English speaker, familiar with the subject area, or a professional editing service, before it can be accepted.

---

## Round 0.4 · Minor Revisions

Dear Authors,

While, as in the previous round, this paper is scientifically ready to be accepted, the English remains problematic in spots. Please either find a copyediting service or a colleague who can bring the work up to international standards. Failure to do so delays final acceptance.

---

## Round 0.5 · accepted · Accept

Many thanks for improving the English of your manuscript. Thank you for submitting your work to this journal.